# How does date-rounding affect phylodynamic inference for public health?

**Leo A. Featherstone**[iD][1,2,3]*, **Danielle J. Ingle**[2], **Wytamma Wirth**[2,4], **Sebastian Duchene**[2,5]

**1** Research School of Biology, Australian National University, Canberra, Australian Capital Territory, Australia, **2** Department of Microbiology and Immunology at the Peter Doherty Institute for Infection and Immunity, University of Melbourne, Melbourne, Victoria, Australia, **3** The Kirby Institute, UNSW Sydney, Sydney, NSW, Australia, **4** Centre for Pathogen Genomics, University of Melbourne, Melbourne, Victoria, Australia, **5** DEMI unit, Department of Computational Biology, Institut Pasteur, Paris, France

* leo.featherstone@unimelb.edu.au

**Data availability statement:** All of the code and date required to reproduce the simulation

## Abstract

Phylodynamic analyses infer epidemiological parameters from pathogen genome sequences for enhanced genomic surveillance in public health. Pathogen genome sequences and their associated sampling dates are the essential data in every analysis. However, sampling dates are usually associated with hospitalisation or testing and can sometimes be used to identify individual patients, posing a threat to patient confidentiality. To lower this risk, sampling dates are often given with reduced date-resolution to the month or year, which can potentially bias inference. Here, we introduce a practical guideline on when date-rounding biases the inference of epidemiologically important parameters across a diverse range of empirical and simulated datasets. We show that the direction of bias varies for different parameters, datasets, and tree priors, while compounding with lower date-resolution and higher substitution rates. We also find that bias decreases for datasets with longer sampling intervals, implying that our guideline is most applicable to emerging datasets. We conclude by discussing future solutions that prioritise patient confidentiality and propose a method for safer sharing of sampling dates that translates them them uniformly by a random number.

## Author summary

Phylodynamic analyses estimate epidemiological parameters using pathogen genome sequences and offer insight for public health. The essential data in every analysis are genome sequences, which allow measurement of evolutionary divergence, and their associated sampling times, which allow evolutionary divergence to be modelled as a rate over time. However, the sampling times of pathogen genome sequences are frequently associated with hospitalisation and can be used to identify particular patients. As a result, sampling times are often shared between public health labs and phylodynamics practitioners with reduced date resolution to protect patient identity (such as to the month or year). Using real-world data and a matching simulation study, we emulate the

study, empirical analyses, and figures/tables are available at
https://github.com/LeoFeatherstone/pdp

**Funding:** This work received funding from: the Inception program (Investissement d'Avenir grant ANR-16-CONV-0005 awarded to SD), the Australian National Health and Medical Research Council (2017284 awarded to SD), and the Australian Research Council (FT220100629 awarded to SD). SD received a salary from the Inception program and the Australian Research Council. LAF received a salary from the National Health and Medical Research Council and Australian Research Council (DP230102424). DJI received a salary from, the National Health and Medical Research Council (GNT1195210). WW was partially supported by a Chan Zuckerberg Initiative Essential Open Source Software for Science grant EOSS6-0000000637. The funders had no role in study design, data collection and analysis, decision to publish or preparation of the manuscript.

**Competing interests:** The authors have declared that no competing interests exist.

effects of date rounding on phylodynamic inference to characterise how reduced date resolution introduces error into inference. We find that error arises where sampling dates are given at a resolution less than the average amount of time it takes for a pathogen to accrue one substitution. We find that this relationship is useful for predicting biased estimation for datasets reflecting short term sampling. We conclude by discussing how accurate sampling dates can be shared in a way that preserves both patient identity and accuracy

## Introduction

Phylodynamics is commonly used to estimate the parameters of viral spread with increasing application to bacteria. It allows estimation of important epidemiological parameters including rates of transmission, the age of outbreaks, rates of spatial advance, and the prevalence of variants of concern [1–4]. It is applicable across the scales of transmission from the pandemic and epidemic scales, such as for SARS-CoV-2 and Ebola virus [5,6], to long-term bacterial transmission such as in *Salmonella enterica* and *Klebsiella pneumoniae*. Phylodynamic analyses are most useful where temporal and spatial records of transmission are sparse, using genomic information to help fill in the gaps.

The basis of all phylodynamic inference is that epidemiological spread leaves a trace in the form of substitutions in pathogen genomes that can be used to reconstruct transmission histories. Pathogen populations meeting this assumption are said to be 'measurably evolving populations' [7,8]. In accordance, phylodynamics uses a combination of genome sequences and associated sampling dates to leverage measurable evolution and infer temporally explicit parameters of transmission and pathogen demography.

Ideal phylodynamic datasets should include precise sampling dates alongside genome sequences [9], but sampling dates necessarily carry over sensitive information about times of hospitalisation, testing, or treatment than can be used to identify individual patients. This can pose an unacceptable risk for patient confidentiality. In some cases, sampling dates or dates of admission are even available for purchase or have allowed identification for a majority of patients in a given record [10]. In acknowledgement of this risk, [11] suggest that Expert Determination govern whether sampling dates be released alongside genome sequences, and the resolution to which they are disclosed (day, month, year). Essentially, this approach involves an expert opinion on whether information is safe to release on a case-by-case basis.

From a phylodynamic point of view, sampling dates with reduced resolution are usable. Uncertainty in sampling dates can be accommodated in Bayesian inference [12], but such an approach is only effective when samples with uncertain dates comprise a small proportion of the total data [13].

The most common technique for incorporating data with a majority of uncertain sampling dates is to assume that sampling occurred at the middle of the uncertainty range, such as all samples from 2020 being assigned 15 June 2020. Other approaches would include sampling a random day within 2020 using a probability distribution over the duration of 2020 for each sample. Both approaches introduce a degree of noise, which may cause bias because sampling dates often drive phylodynamic inference [14–16]. Understanding this bias has practical significance, as there are many examples of phylodynamic analyses conducted with reduced date resolution for a diverse array of pathogens. These include viral pathogens such as Rabies virus, Enterovirus, SARS-CoV-2, Dengue virus [17–20], and bacterial pathogens, such as *Klebsiella pneumoniae*, *Streptococcus pneumoniae*, and *Mycobacterium tuberculosis* [21–23].

Precision in sampling dates is also relevant to the design and curation of pathogen sequence databases because sampling dates are often considered as metadata, and thus recorded inconsistently throughout repositories [24]. For example, as of early September 2024, there were roughly 19.9M SARS-CoV-2 genome sequences available on GISAID with roughly 2.4% (382K) of these having incomplete date information, where sampling dates are absent or only given to the month or year. In other words, roughly 1 in 50 sequences lacked clear date resolution, reflecting global inconsistency in SARS-CoV-2 sampling time records.

In recognition of this issue, we characterised the conditions under which biases arise from reduced date resolution in phylodynamic inference. We analysed four empirical datasets of SARS-CoV-2, H1N1 Influenza, *M. tuberculosis*, *Staphylococcus aureus*, and conducted a simulation study with parameters corresponding to each empirical dataset. We also included a supplementary H3N2 influenza dataset. These pathogens are key examples of candidates for genome surveillance, with SARS-CoV-2, H1N1, and H3N2 having caused pandemics and *S. aureus* and *M. tuberculosis* being global priority pathogens [25]. These data also have diverse infectious periods and molecular evolutionary rates, thus providing a broad representation of phylodynamics' applicability to pathogens presenting human-health threats. For each empirical and simulated dataset, we studied the bias in estimated epidemiological parameters across treatments with sampling dates rounded to the day, month, or year. For example, 2021-10-11 would be specified as 2021-10-15 when rounding to the month and 2021-06-15 when the month and day are not provided.

We focused on inference of the reproductive number ($R_0$ or $R_e$ for the basic and effective reproductive number, respectively), defined as the average number of secondary infections stemming from an individual case (reviewed by [1,3,26]), the time to the most recent common ancestor (tMRCA), and the substitution rate (substitutions per site per year) in each dataset. Together, these parameters span much of the insight that phylodynamics offers through inferring when an outbreak started and how fast it proceeded. The evolutionary rate is also the central parameter relating evolutionary time to epidemiological time, so any resulting bias in this parameter is expected to have a pervasive effect throughout each phylodynamic model.

We hypothesised that reduced date resolution causes bias that compounds where the uncertainty in dates exceeds the average time for a substitution to arise in a given pathogen. That is, the point from which substitution events are conflated in time. We visualise the relationship between date resolution and average substitution time in Fig 1. For example, H1N1 influenza virus accumulates substitutions at a rate of about $4 \times 10^{-3}$ subs/site/year [27]. With a genome length of 13,158bp, we then expect roughly one substitution to accrue per week. Therefore, rounding dates to the month or year conflates molecular evolution in time and biases inference. Based on this, we expected the SARS-CoV-2 and H1N1 datasets to exhibit bias from month resolution onwards, the *S. aureus* dataset to exhibit bias at year resolution, and the *M. tuberculosis* dataset to not display bias up to and including year resolution (See Table 1 for average substitution times). Throughout this manuscript, we refer to bias where we recover error in estimated parameters with a consistent direction among replicates in our simulation study (consistently over- or underestimating). We do not chiefly consider the variance in posterior distributions of estimated parameters, but discuss this point in the results.

Our results across the simulation study and analyses of empirical data support using the average substitution time as a rough threshold for when date-rounding causes bias. We also consider factors that modulate the extent of bias, in particular noting that it declines with longer sampling intervals, and varies in direction between datasets and tree prior. We finish by discussing future solutions that prioritise both patient confidentiality and accurate data sharing for routine phylodynamic analyses in public health.

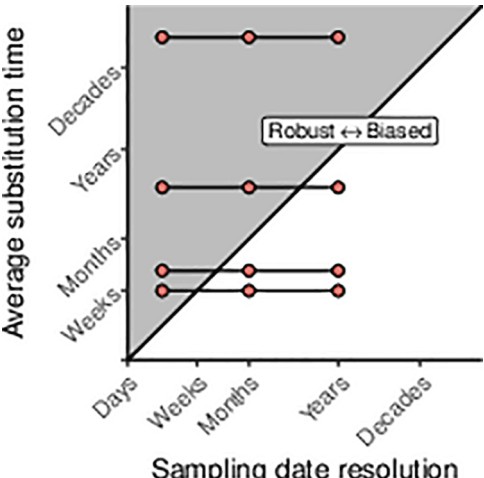

**Fig 1. Graphical representation of the hypothesis.** The average time to accrue one substitution based on a fixed genome size and evolutionary rate, $T_s = [\text{Genome Length (sites)} \times \text{Evolutionary rate (subs/site/yr)}]^{-1}$ against the temporal resolution lost by date-rounding. We hypothesised that and showed that when analyses for a given pathogen round dates to an extent nearing or crossing the diagonal from left to right, biases are induced in $R_e$, tMRCA, and substitution rate. substitution rates are taken from each source for the empirical data. We do not report the numerical axis as this figure is designed to illustrate a concept rather than serve as a reference, in the same spirit as its inspiration in Fig 2 of [8].

**Table 1. Substitution rates and genome length for sequence simulation.**

| Microbe | Substitution Rate (subs/site/yr) | Genome Length | Time/Sub/Genome (yrs) |
|---|---|---|---|
| H1N1 | $4 \times 10^{-3}$ | 13158 | 0.0190 |
| SARS-CoV-2 | $1 \times 10^{-3}$ | 29903 | 0.0334 |
| *S. aureus* | $1 \times 10^{-6}$ | 2900000 | 0.3458 |
| *M. tuberculosis* | $1 \times 10^{-7}$ | 4300000 | 2.3256 |

## Methods

### Overview

Our study is based on four empirical datasets including two viruses, H1N1 influenza and SARS-CoV-2, and two bacterial species, *Staphylococcus aureus* and *Mycobacterium tuberculosis*. We also conducted a simulation study with parameters tailored to each dataset. These data were chosen to span the usual parameter space for substitution rate and sampling duration in phylodynamics for epidemiology (roughly $10^{-3}$-to-$10^{-8}$ (subs/site/yr) for substitution rate and months-to-decades for duration of sampling). We also included a supplementary H3N2 influenza empirical dataset to illustrate the effects of date rounding on longer-term viral datasets.

To assess the effects of date-rounding, we conducted phylodynamic analyses for both the empirical and simulated datasets with sampling dates rounded to the day, month, or year. For example, two samples from 2000-05-29 and 2000-05-02 would both become 2000-05-15 if rounded to the month. We then measured the resulting bias in epidemiologically- or phylodynamically-important parameters: the reproductive number ($R_0$ or $R_e$), substitution rate (subs/site/year), and the tMRCA. The tMRCA gives a measure of the age of the pathogen population driving the outbreak and is often interpreted as the age of the outbreak. We also

consider the tMRCA to facilitate comparison, because there is variability in which phylodynamic models include the length of the root branch in the age of the outbreak [28].

The viral datasets consist of samples from the 2009 H1N1 pandemic (n=161) from [27], and a cluster of early SARS-CoV-2 cases from Victoria, Australia in 2020 (n = 112) [29]. The bacterial datasets consist of *S. aureus*, with 104 samples from New York sampled over ≈2 years [30–32], and 30 *M. tuberculosis* samples from an ≈25 year outbreak studied by [33]. These data were chosen because they encompass a diversity of epidemiological dynamics, timescales, and variable substitution rates.

## Simulation study

We simulated outbreaks as birth-death sampling processes using the ReMaster package in BEAST v2.7.6 [34,35]. Simulations employed four parameter settings corresponding to each empirical dataset (Table 2), with 100 replicates of each. All parameter sets include a proportion of sequenced cases ($p$), outbreak duration ($T$), a 'becoming un-infectious' rate ($\delta = \frac{1}{\text{Duration of infection}}$), and transmission rates via reproductive numbers. We matched the values of each parameter to those in the originating literature (Table 2). We fixed the sequencing proportion in the H1N1 simulations by dividing the sample size ($n = 161$) and the cumulative number of North American cases over the empirical data's sampling interval, resulting in the order of 1% [36].

For simulations corresponding the viral datasets, transmission was modelled via $R_0$, the average number of secondary infections (assuming a fully susceptible population). For those corresponding to the bacterial datasets, we allowed the effective reproductive numbers to vary over two intervals ($R_{e_1}$ and $R_{e_2}$ respectively). For the *S. aureus* setting, the change time for $R_e$ was set at $t = 22$ with the sequencing proportion ($p$) also set to zero before this time to replicate the sampling effort in the empirical dataset. For the *M. tuberculosis* dataset, the change time was fixed at halfway through simulations ($t = 12.5$) with one fixed sequencing proportion throughout.

Simulations generated a total of 400 outbreaks which we then used to simulate sequences data under a Jukes-Cantor model using Seq-Gen v1.3.4 [37] with fixed substitution rates (Table 1). We chose a simple substitution model to reduce parameter space and because substitution model mismatch has been widely explored elsewhere (e.g. [38]).

We then analysed each of the 400 simulated datasets under three date resolutions (day, month, and year), and two tree priors: the birth-death [28] and coalescent with exponential growth, referred to hereon as the 'coalescent exponential' [39]. This yielded 1800 analyses

**Table 2. Parameter sets for the simulation study corresponding to each empirical dataset.** $\delta$ is the 'becoming un-infectious' rate, which is the reciprocal of the duration of infection in units of years$^{-1}$. $R_0$ is the basic reproductive number, describing the average number of secondary infections arising at the beginning of an outbreak where the susceptible population is greatest. $R_{e_\bullet}$ refers to the effective reproductive number over two successive intervals of an outbreak as the susceptible population varies. $p$ is the proportion of sequenced cases. $T$ is the duration of the outbreak.

| Microbe | $\delta$ (yrs$^{-1}$) | $R_0$ | $R_{e_1}$ | $R_{e_2}$ | $p$ | $T$ (yrs) | Source |
|---------|------------|-------|-----------|-----------|-----|-----------|--------|
| H1N1 | 91.31 | 1.3 | - | - | 0.015 | 0.25 | [27] |
| SARS-CoV-2 | 36.56 | 2.5 | - | - | 0.80 | 0.16 | [29] |
| *S. aureus* | 0.93 | - | 2.0 | 1.0 | 0.2$^\dagger$ | 25 | [30] |
| | | | | | | | [31] |
| *M. tuberculosis* | 0.125 | - | 2.0 | 1.10 | 0.08 | 25.0 | [33] |

$^\dagger$ $p$ was set to zero before $T = 22$

in total (1200 for the birth-death and 600 for the coalescent exponential). We used identical model specifications and prior distributions as for the corresponding empirical datasets. We ran each MCMC chain for $5 \times 18^8$ steps, sampling every $10^{4\text{th}}$ step and discarding the first 50% as burnin. We then discarded all analyses that did not have effective sample sizes ($ESS$) of at least 200 ($ESS \geq 200$) for every parameter, leaving a total of 1670 replicates incorporated in our results.

## Empirical data

We conducted Bayesian phylodynamic analyses using a birth-death skyline tree prior in BEAST v2.7.6 for all datasets [35]. We also fit a coalescent exponential tree prior to the viral datasets. We did not fit the coalescent exponential to the bacterial datasets because they capture transmission beyond the exponential phase, which would therefore result in model misspecification. We sampled from the posterior distribution using Markov chain Monte Carlo (MCMC), with $5 \times 10^7$ steps ($1 \times 10^7$ for SARS-CoV-2 data), sampling every $10^4$ steps, and discarding the initial 10% as burnin. We assessed sufficient sampling from the stationary distribution by ensuring $ESS \geq 200$ for all parameters and likelihoods.

**H1N1.** The H1N1 data consist of 161 samples from North America during the 2009 H1N1 influenza virus pandemic, previously analysed by [27]. Samples originate from April to September 2009 and provide an example of a rapidly evolving pathogen sparsely sequenced during an emerging outbreak.

Under the birth-death model, we placed a Lognormal($\mu = 0, \sigma = 1$) prior on $R_0$, $\beta(1,1)$ prior on $p$, and fixed the becoming-uninfectious rate to ($\delta = 91 \ years^{-1}$), corresponding to a four-day duration of infection. We also placed an improper ($U(0, \infty)$) prior on the age of the outbreak and a Gamma(shape = 2, rate = 400) prior on the substitution rate.

Under the coalescent exponential, we placed a Laplace($\mu = 0$, scale = 100) prior on the growth rate, which was later transformed to $R_0$ ($R_0 = rD + 1$ where $r$ is the growth rate and $D$ is the duration of infection). We also placed an improper prior ($\frac{1}{x}$) on the effective population size, which is the maximally uninformative Jeffrey's prior for coalescent intervals [40]. We otherwise included the same priors as for the birth-death.

**SARS-CoV-2.** The SARS-CoV-2 data consist of 112 samples from a densely sequenced transmission cluster from Victoria, Australia over late July to mid September 2020 [29]. These data are similar to the H1N1 datasets in presenting a quickly evolving viral pathogen, but differ in that a high proportion of cases were sequenced.

Under the birth-death, we placed a Lognormal(mean = 1, sd = 1.25) prior on $R_0$ and an Inv-Gamma($\alpha = 5.807, \beta = 346.020$) prior on the becoming-uninfectious rate ($\delta$). The sampling proportion was fixed to $p = 0.8$ since the target was to sequence every known SARS-CoV-2 case in Victoria at this stage of the pandemic, with a roughly 20% sequencing failure rate. We also placed an Exp(mean = 0.019) prior on the origin, corresponding to a lag of up to one week between the index case and the first putative transmission event. In this case, the origin parameter corresponds to the length of the root branch. In the results we still report the age of the outbreak as the tMRCA for consistency with the other datasets. Lastly, we placed a Gamma(shape = 2, rate = 2000) prior on the substitution rate.

Under the coalescent exponential, we placed an improper prior ($\frac{1}{x}$) on the effective population size and a Laplace($\mu = 0.01$, scale = 0.5) prior on the growth rate. Other parameters were given the same priors as under the birth-death. Note that the coalescent exponential is not a natural choice of tree prior for the SARS-CoV-2 data because of its very high sequencing proportion [41]. We nevertheless include it for the SARS-CoV-2 data to provide

a comparison to the coalescent exponential for the H1N1 data, as well as an example of how data-rounding may exacerbate error in conjunction with poorly fitting models. The model's poor fit is reflected later in the results.

**H3N2.** We included a supplementary H3N2 dataset to assess the effects of date rounding for a viral dataset with longer term sampling. Using the multi type birth-death model, we analysed a 60 H3N2 influenza samples taken from 2000 to 2005 in Hong Kong and New Zealand, with demes corresponding to each location [42]. The data are a subset of those originally used in [43], and is also available from the structured birth-death 'Taming the BEAST' tutorial [44].

We placed a Lognormal($\mu = 0, \sigma = 1$) prior on $R_0$ for both demes and fixed the becoming un-infectious rate at $\delta = 71$, corresponding to a roughly 5-day duration of infection. We also placed a Lognormal($\mu = 0.001, \sigma = 1.25$) prior on the substitution rate and sampling probability ($p$). The sampling probability was fixed to zero shortly before the oldest sample in all date treatments to reflect no sampling effort prior to this date.

***Staphylococcus aureus.*** The *S. aureus* dataset originates from [32] and we analysed a subset of the data later analysed in [30] and [31]. It consists of a single nucleotide polymorphism (SNP) alignment of 104 sequenced isolates sampled in New York from 2009 to 2011. Populations growth is understood to have been driven by $\beta$-lactam antibiotic use beginning in the 1980s. These data therefore provide a comparison to the *M. tuberculosis* dataset in a briefer sampling span from an outbreak of similar duration.

To accommodate changing transmission dynamics, we included two intervals for $R_e$ with a Lognormal($\mu = 0, \sigma = 1$) prior on each. We also placed a $\beta(1, 1)$ prior on the sampling proportion, which was otherwise fixed to 0 before the first sample to capture the lag in sampling. We also placed a $U(0, 1000)$ prior on the origin, and fixed the becoming un-infectious rate at $\delta = 0.93$, corresponding to a nearly year-long duration of infection following [31].

***Mycobacterium tuberculosis.*** The *M. tuberculosis* dataset consists of 36 sequenced isolates from a retrospectively recognised outbreak in California, USA, that originated in the Wat Tham Krabok refugee camp in Thailand. The data were originally analysed using the birth-death tree prior by [33]. We applied the same prior configurations as [33], with the exception of including two intervals for $R_e$ and fitting a strict molecular clock with a Gamma(shape = 0.001, rate = 1000.0) prior.

## Results

### Simulation study

The viral simulation conditions (i.e. SARS-CoV-2 and H1N1) display the greatest bias in mean posterior estimates of substitution rate, tMRCA, and reproductive number with decreasing date resolution (Fig 2A–2C). The *S. aureus* simulations exhibit a similar trend when rounding dates to the year, although with lesser error than for the viral simulations. The *M. tuberculosis* condition is effectively inert to decreasing date resolution, with mean posterior estimates for each parameter of interest remaining consistent across date resolution (day to year). The *S. aureus* data provide an important intermediate case in that estimates of each parameter change when transitioning from month to year resolution (see crossing of lines from month to year resolution in the *S. aureus* column of Fig 2). These trends are in agreement with the hypothesis of bias arising where date resolution is less than the average time substitution time. This occurs because date-rounding compresses divergent sequences in time, driving a signal for higher rates of substitution and transmission local to each temporal cluster of sampling dates. This effect is less pronounced in the bacterial simulation conditions

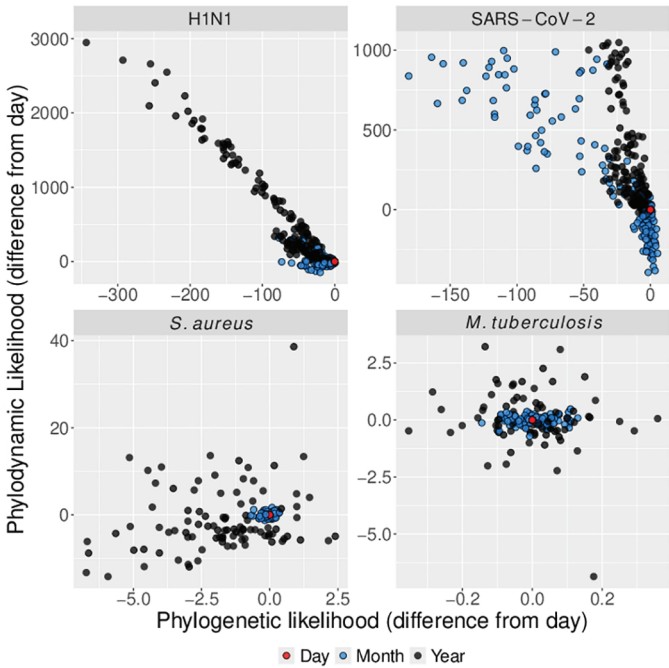

**Fig 2. Mean posterior estimates for parameters of interest for each simulated dataset varying across date resolution.** Individual lines track mean posterior estimates for each simulated dataset and boxplots are given to summarise the spread and direction of bias across all simulated datasets at each date resolution. Rows correspond to individual parameters, columns correspond to simulation conditions (underlying parameters matching each empirical dataset), and colour corresponds to tree prior or reproductive number interval. Dashed horizontal lines correspond to the true value under which each dataset was simulated. (**A**) Mean posterior substitution rate across simulation scenarios. (**B**) Mean posterior tMRCA, a measure of the age of the population driving the outbreak. (**C**) Mean posterior reproductive number.

relative to the viral conditions, because the date resolution lost is a smaller fraction of the effective substitution time (average time to until substitution is $\approx 4$ months and $\approx 28$ months for *S. aureus* and *M. tuberculosis* conditions respectively, Table 1). In other words, the bacterial sequences clustered in time were on average less divergent than for the viral data, which is biologically realistic given that bacteria tend to accrue substitutions more slowly than viruses. There are also notable deviations from these general trends across date resolutions, simulation conditions, and tree priors that we attribute to the duration of the sampling intervals below. The posterior variance for each estimated parameter also tended to increase with decreasing date resolution, demonstrating that it results in increased uncertainty as well as bias (Fig A in S1 File).

The coalescent exponential shows overall downwards bias in the substitution rate for the SARS-CoV-2 and H1N1 treatments at month resolution, while the birth-death exhibits upwards bias. Since the sampling dates for each viral dataset are distributed over three months, date-rounding compresses samples within a month to one time, simultaneously increasing the time between samples across months and driving a signal for lower transmission and substitution rates between months. The different phylodynamic likelihood functions for each tree prior respond differently to this warped distribution of diversity over time with the coalescent exponential placing more weight on decreased rates of substitution rates

while the birth-death favoured an increase. This can be explained by the birth-death drawing signal for increased transmission among coincident sampling dates within each month, while the coalescent exponential instead conditions on sampling dates [16]. At year resolution there is also lower bias in estimates of substitution rate for the coalescent exponential than the birth-death, however both models estimate upwards-biased substitution rates as year resolution. This is probably because year resolution clusters all sampling dates to a single time, meaning a highly inflated rate of substitution is needed to model the artificial burst in diversity at one time for both tree priors. Fig B in S1 File shows sampling dates compressed in time across date resolution for posterior trees. For all viral simulation conditions, the mean posterior tMRCA of each outbreak shifts inversely to the substitution rate. This is the result of a well understood relationship among phylodynamic models where higher rates of evolution suggest shorter periods of evolution.

The reproductive number for each viral dataset ($R_0$) also changes markedly with decreasing date resolution under the birth-death, but not under the coalescent. For the birth-death, this is in agreement with temporal clustering of samples driving a signal for higher transmission rates. Conversely, estimates under the coalescent exponential remain near-identical at month resolution, which is again due to its conditioning on sampling dates. Estimates of $R_0$ for the SARS-CoV-2 settings under the coalescent exponential are also heavily biased downwards. This is probably due to high sequencing proportions violating the assumption of low sampling under the coalescent, thus leading to poorly fitting model in the first place.

The *S. aureus* condition yields consistent estimates of substitution rate, tMRCA, and reproductive number ($R_e$ in this case) when days are rounded to the month (Fig 2 *S. aureus* column). At year resolution the posterior substitution rate appears biased downwards. This can be explained by the two year sampling duration of the *S. aureus* condition, such that samples rounded to the year will be on average further apart in time than if dates are given to the month or day (Fig B in S1 File). This spacing of diversity in time likely drives the signal for lower substitution rates and an older outbreak in turn. There is no clear pattern in the direction of bias for $R_{e_1}$ and $R_{e_2}$ at year resolution, though estimates deviate from those at month and day resolution. Estimates for $R_{e_1}$ are also overall lower than their true value of 2.0, and this is attributable to inconsistent sampling over the duration of the outbreak which was previously demonstrated for other datasets with late sampling in [15].

The *M. tuberculosis* simulation condition effectively acts as a control, since it appears inert to date-rounding. This is expected because this dataset reflects longer simulation time, with temporal clustering less likely to inflate $R_e$, and an average substitution time is longer than a year. As such, even rounding to the year is unlikely to drive a signal for increased evolutionary rate or a more recent origin time.

Phylodynamic and phylogenetic terms from the total posterior likelihood also vary with decreasing date resolution, with deviation increasing with decreasing date resolution from month and year (Fig C in S1 File). This verifies that altered date resolution affects the likelihood manifold of each analysis, which is reflected in the different trends of bias in each parameter of interest.

## Empirical results

Broadly, analyses of the empirical datasets reproduce the patterns of bias in the simulation study (Fig 3). That is, the reproductive number increases with decreasing date resolution along with an increase in the substitution rate and corresponding decrease in the tMRCA. There are a few exceptions to this trend that we consider below and which we again attribute

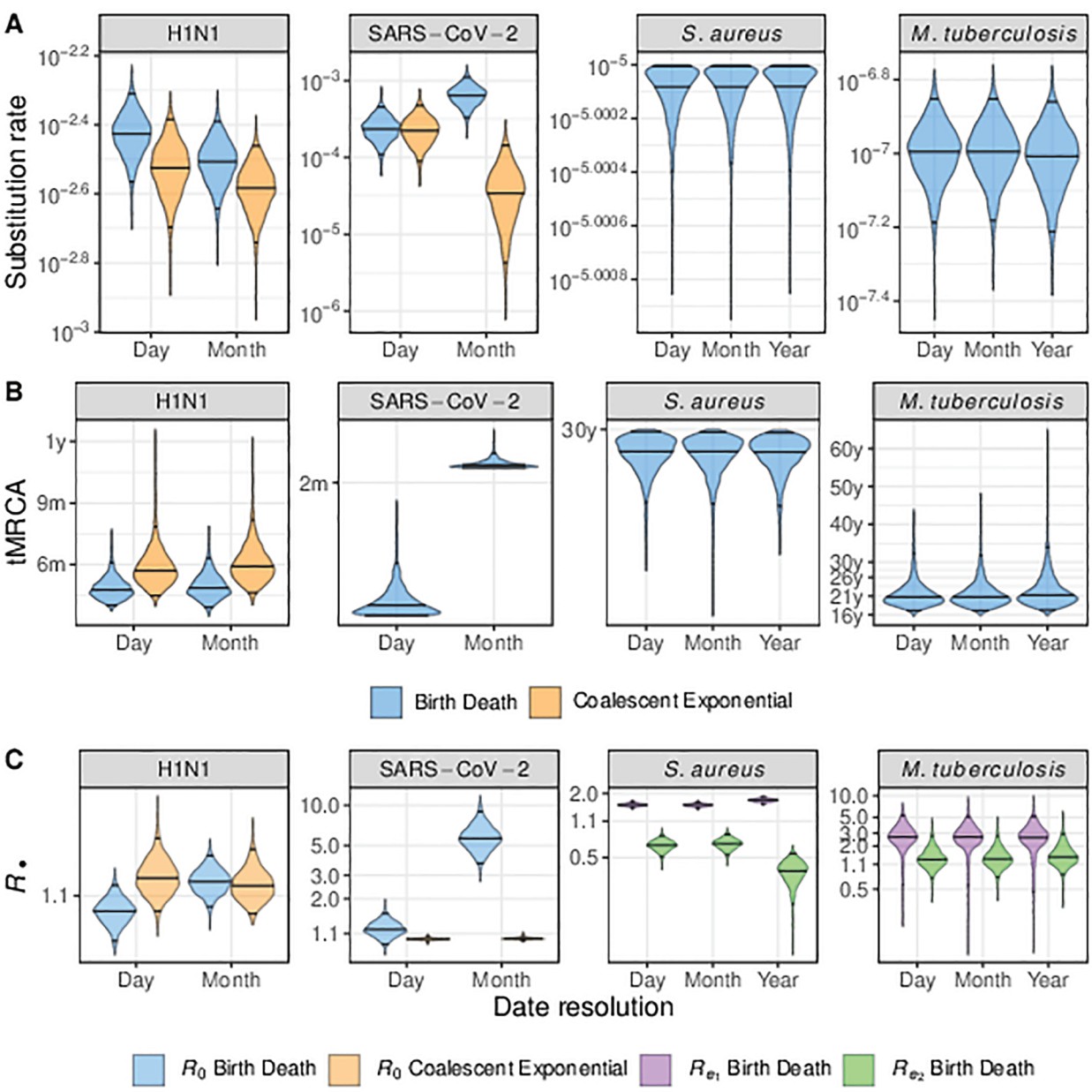

**Fig 3. Posterior distributions for parameters of interest estimated for each empirical dataset.** Date resolution is given on the horizontal axis and colour denotes tree prior. Horizontal lines in each violin plot denote the median and 95% highest posterior density interval. Estimates for viral datasets at year-resolution are omitted because results deviate by implausible orders of magnitude due to sampling dates being identical. (**A**) Posterior substitution rate across date resolutions. (**B**) Posterior tMRCA in units of months (m) or years (y). Posterior tMRCA for SARS-CoV-2 data under the coalescent exponential are omitted because they are effectively flat over zero to twenty years or more, obscuring the other distributions. The figure with the full range is available at online. (**C**) Posterior reproductive number on a log-transformed axis.

to the difference between simulated and empirical sampling time distributions. Supplementary analysis of the H3N2 data further demonstrates that bias due to date-rounding lessens with longer sampling windows (Fig D in S1 File).

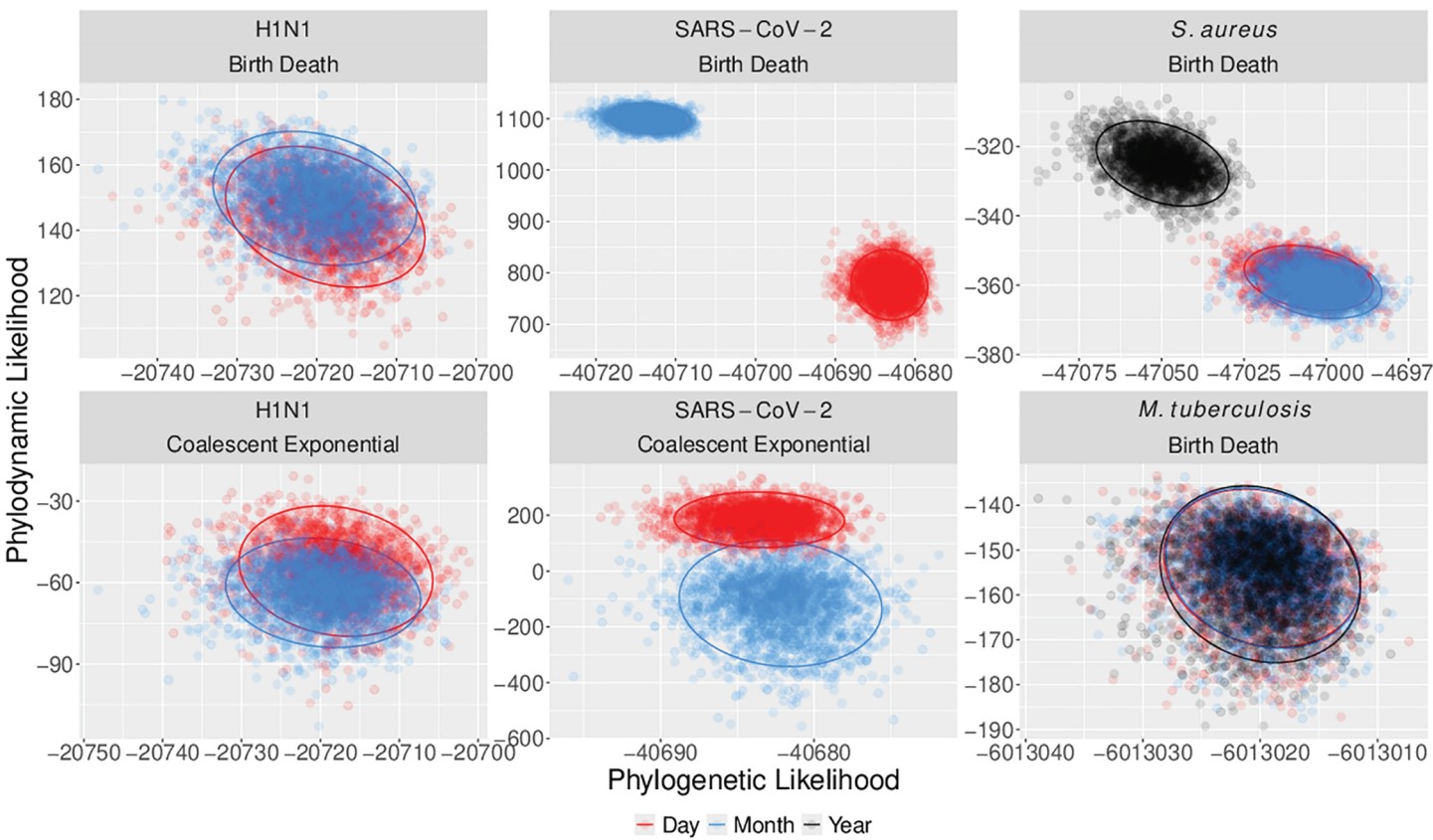

**Fig 4. Posterior phylodynamic likelihood against phylogenetic likelihood for each combination of empirical dataset and with colour corresponding to date resolution.** Ellipses surround the 95% highest posterior density region for each posterior. Both phylodynamic and phylogenetic likelihoods diverge between day and month resolution for the SARS-CoV-2 datasets. Divergence is lesser for the H1N1 dataset, reflecting a longer sampling span. Year resolution differs from month and day for the *S. aureus* data. Posterior likelihoods all coincide for *M. tuberculosis*.

Phylodynamic and phylogenetic likelihoods also appear to diverge when date resolution is less than the average substitution time (Fig 4). The SARS-CoV-2 data show a clear separation in likelihoods from day to month resolution for both tree priors. Likelihoods also separate at year resolution for the *S. aureus* data and remain nearly identical for *M. tuberculosis*. However, likelihoods overlap considerably at day and month resolution for H1N1 under both tree priors, which differs from expectation. This is probably due to longer sampling duration, compared to the SARS-CoV-2 data, lessening the effects of data rounding. Together, these movements in likelihood distributions support the hypothesis that date resolution less than the average substitution time leads to bias, with this effect reducing for longer sampling duration.

**H1N1.** Mean posterior $R_0$ increases from day to month resolution for the birth-death (1.07 to 1.12), yet remains near-identical for the coalescent exponential (1.13 to 1.12) (Table A in S1 File). The mean posterior substitution rate also decreases for both tree priors across day to month resolution ($3.8 \times 10^{-3}$ to $3.1 \times 10^{-3}$ and $3.01 \times 10^{-3}$ to $2.62 \times 10^{-3}$ for the birth-death and coalescent exponential respectively) (Table A in S1 File). The posterior tMRCA also differs between tree priors mirroring substitution rate, with a decrease from date to month

resolution for the birth-death and an increase for the coalescent exponential. For the coalescent exponential, we can attribute the decrease in reproductive number and substitution rate from day to month resolution to samples being spread further in time (Fig E in S1 File), which drives the signal for and an older outbreak and lower transmission rates. While the same is true for the sampling distribution under the birth-death, the additional information it draws from identical sampling dates as month-resolution likely inflates the mean posterior reproductive number despite the signal for a lower substitution rate and older outbreak.

**SARS-CoV-2.** Under the birth-death, the SARS-CoV-2 dataset behaves as expected, with an increase in posterior $R_0$ from day to month rounding. In particular, rounding to the month results in a high, yet plausible value of $R_0$ = 5.972 (Table A in S1 File). Under the coalescent exponential, the mean posterior $R_0$ remains near identical across day to month treatments (1.00 to 1.01 respectively). We again note that the coalescent exponential is included for completeness for the SARS-CoV-2 dataset, but is not an appropriate choice of model in practice due to the near complete-sequencing of the original transmission cluster. Thus, poor model-fit is probably the cause of unrealistic estimates of $R_0$.

The mean posterior substitution rate under the birth-death increases over two-fold when rounding to the month ($2.47 \times 10^{-4}$ to $6.56 \times 10^{-4}$, Table A in S1 File). Mean posterior tMRCA also increases from 0.15 years to 0.17 years from day to month, which contradicts the expectation of a decreased estimate of tMRCA under date-rounding. We again attribute these differences to the distribution of the empirical sampling dates under date-rounding. Sampling for the SARS-CoV-2 dataset mainly occurred over August to September 2020, with most August samples originating later in the month (Fig E in S1 File). Rounding to $15^{th}$ of August therefore made these samples appear older in time and likely contributed to the older origin under month-rounding.

**H3N2.** The H3N2 data were more robust to date rounding than the H1N1 and SARS-CoV-2 datasets, which were sampled over shorter time frames (Fig D in S1 File). Posterior $R_0$ for Hong Kong, from which the majority of samples originate, remains nearly identical across date resolutions (mean posterior values of 1.00 for each date resolution). Posterior $R_0$ for the New Zealand deme also remains near-identical at day and month resolution, before increasing in uncertainty at year-resolution (mean values of 0.99, 0.99, and 0.97 respectively). The mean posterior substitution rate decreases at year resolution ($3.13 \times 10^{-3}$, $3.11 \times 10^{-3}$, $2.86 \times 10^{-3}$), while the tMRCA increases in step with each reduction in date resolution from day, to month, to year (6.87, 6.93, 7.04). The relative stability of posterior estimates across date resolutions for the H3N2 dataset differs from the bias induced in the H1N1 and SARS-CoV-2 datasets, supporting the notion that shorter sampling spans exacerbates bias due to date-rounding.

**S. aureus.** For $R_{e_1}$, the S. aureus dataset recapitulated the simulation study with month rounding having a minimal effect, but year rounding inducing an upwards bias (mean values of 1.57, 1.56, 1.73 respectively)(Fig 3, Table A in S1 File). $R_{e_2}$ displays a similar pattern with consistent estimates at day and month-rounding before a reduction at year rounding (0.66, 0.67, and 0.37 respectively). This result is consistent with the estimates an initial increase in growth rate in previous analyses of the dataset [31].

Mean posterior substitution rate and tMRCA remain identical across date resolutions ($10^{-5}$ subs/site/year and a tMRCA of 30 years), despite the change in reproductive numbers at year rounding. This is surprising given the change in posterior phylodynamic and phylogenetic likelihoods (Fig 4), and highlights that date-rounding can perturb the likelihood without predictable changes in parameters of epidemiological significance.

*M. tuberculosis.* The *M. tuberculosis* data recapitulate the outcome of the simulation study in being robust to date-rounding. Posterior substitution rates and outbreak ages remain consistent across decreasing date resolution ($1.02 \times 10^{-7}$, $1.02 \times 10^{-7}$, $9.86 \times 10^{-8}$ (subs/site/time) and 21.7, 21.7, and 22.5 years respectively) (Table A in S1 File, Fig 3). We also infer that $R_{e_1} > R_{e_s}$ across date-rounding conditions, coinciding with an earlier burst of transmission in agreement with [33]. However, $R_{e_1}$ decreases slightly with date-rounding (mean posterior estimates of 2.77, 2.74, 2.66 for day, month and year rounding)(Table A in S1 File), while $R_{e_2}$ increases (1.4, 1.41, 1.53 from day to year rounding). This was likely caused by the higher number of samples in the second sampling interval, from roughly 2002 to 2010, such that compressing sampling dates drive drove a signal for higher transmission in the second interval with longer periods between sampling in the first interval at year resolution. Again, this shows that distribution of sampling dates for empirical data, which is largely unpredictable, modulates the effects of date-rounding.

## Discussion

The results of the simulation study and analyses of empirical data support our hypothesis that phylodynamic inference is most biased where the temporal resolution lost in date rounding exceeds the average time for one substitution to arise. In the both the simulation study and empirical analyses, the viral datasets (H1N1 and SARS-CoV-2) display the greatest bias in mean posterior reproductive number, substitution rate, and tMRCA when rounding to the month or year, with the average substitution time being less than one month in both simulation conditions. The *S. aureus* data provide an intermediate case, with estimated parameters displaying bias when rounding dates to the year (average substitution time between the order of months to a year). Lastly, the *M. tuberculosis* data also provide supporting evidence in not displaying any notable bias between estimates at day, month, or year date-rounding. This is expected because the average substitution time is longer than a year in all *M. tuberculosis* analyses.

We therefore propose the average substitution time as a rough practical threshold after which genomic epidemiologists can invariably expect date-rounding to distort inference. Genomic epidemiologists can make this assessment by calculating the average substitution time, $T_s$, as $T_s = [\text{Genome Length (sites)} \times \text{Evolutionary rate (subs/site/yr)}]^{-1}$ and checking whether $T_s < \frac{1}{12}$ (indicating substitutions arising faster than monthly) when justifying rounding to the day, or $T_s < 1$ (substitutions arising more than yearly) when justifying rounding to the month. In the more general terms, we propose that date rounding is problematic for fast-evolving RNA viruses, such as in the H1N1 and SARS-CoV-2 datasets. We urge others uploading data to repositories to include dates to the day where possible. This will increase the added-value of phylodynamic analysis for future infectious disease threats. Rounding to the year is sufficient for slowly evolving bacteria such as *M. tuberculosis*. We suggest case-by-case assessment for pathogens with intermediate average substitution times, such as the *S. aureus* herein and other faster-evolving bacteria including *Streptococcus*, multi-drug resistant *Escherichia coli*, or *Klebsiella pneumoniae* [45–47]. In the specific cases of *S. aureus* and other high disease-burden bacteria with asymptomatic and/or community carriage, we suggest preserving dates as much as possible to recover maximal information given the additional work that is often dedicated to screening samples. Finally, we note that genome samples with or without rounded dates reflect considerable efforts in the field to collect and process samples. In the case where only low-resolution dates are available, we do not discourage phylodynamic analyses, but instead encourage additional analyses to test the effects of rounded dates, such as by including priors on sampling ranges.

We strongly emphasise that this proposal is a rough guideline lacking rigorous mathematical derivation. Any degree of date-rounding may alter likelihood and parameter estimation in phylodynamic analyses. Other factors such as as the length of the sampling window, distribution of sampling dates, and choice of tree prior also affect the direction and severity of bias when rounding dates.

Longer sampling intervals also reduce bias due to date-rounding. This is demonstrated by supplementary analysis of the H3N2 data, spanning roughly 6 years. Posterior estimates of $R_0$, tMRCA, and substitution rate vary much less with lower date-resolution than for the other H1N1 and SARS-Cov-2 datasets sampled over three and two months respectively. In turn, the H1N1 dataset also demonstrates lesser bias than the SARS-CoV-2 dataset. This is consistent with previous results for ancient DNA data showing that date-rounding has negligible effects for timescales of millennia or longer, where the sampling span covers the average substitution interval many times [48]. The mitigating effect of sampling duration implies that the hypothesis relating bias to average substitution time and date resolution is of practical significance for emerging datasets sampled over shorter time frames. Conversely, datasets reflecting longer-term sampling are more likely to be robust to date-rounding.

Determining the sampling duration after which inference is sufficiently robust to date-rounding requires case-by-case consideration. The answer will reflect multiple factors including of the pathogen's substitution rate, the number and temporal-distribution of samples, and aspects of inference including the choice of model and priors. For example, although roughly six years of sampling largely eliminated bias at the month resolution for the H3N2 data, a longer time frame would be needed for a bacterial dataset to span an equivalently informative amount of substitutions. Moreover, we note that longer sampling duration improves bias by including more internal branches that drive a signal for lower rates of substitution and transmission, counteracting the opposite signal local to each cluster of sampling times. That is, longer sampling durations introduce a balancing source of error, rather than directly accounting for the inaccuracy in sampling times. Is is therefore possible that extremely long sampling durations in combination with date rounding could exacerbate the opposite bias, inferring erroneously low substitution and transmission rates. Given this potential confounding factor, we suggest that the best solution is prevention wherein the most precise dates that balance patient confidentiality and phylodynamic accuracy are prioritized.

The choice of tree prior also affects bias when rounding dates. For example, the coalescent exponential tended to infer decreased substitution rates while the birth-death favoured increased substitution rates across simulated and empirical viral data. The inverse trend also arose for the tMRCA. This is because the birth-death draws additional information from clustered sampling dates, which serves to elevate rates of substitution and transmission, while the coalescent conditions on these and draws more from the longer duration between sampling dates at month resolution. These patterns emerged even for the coalescent exponential fit to simulated and empirical SARS-CoV-2 data, where it is a poor fit. This highlights that date rounding has a pervasive effect distinct from the effects of a poorly fitting tree prior.

In sum, the results form the simulation study and empirical data show that although date-rounding biases epidemiological estimates in a theoretically predictable directions (upwards for transmission and substitution rates, downwards for tMRCA), the intensity of the bias is difficult to predict and varies with the distribution and span of sampling dates as well as tree prior. We conclude that sufficiently accurate sampling dates are essential where phylodynamic insight is needed to understand infectious disease epidemiology and evolution. There does not appear to be a clear way to adjust for the bias caused otherwise. Accurate sampling dates will be essential for employing phylodynamics amid future infectious disease threats.

We also acknowledge that while accurate sampling dates are essential for reliable phylo-dynamic results, it may pose an unacceptable level of risk to patient confidentiality to release sampling dates. We therefore emphasise the importance of methods that prioritise both patient confidentiality and data transparency and finish by discussing potential future solutions.

## Translating dates by random seeds

The functional component of phylodynamic data are the differences among genome sequences and among dates, rather than their absolute values. It may therefore be possible to protect patient confidentiality while sharing accurate dates by translating dates uniformly by a random number. This would protect the true sampling dates while preserving the relative times between them. For example, if the sampling dates in a dataset of 3 genomes are 2000, 2001 and 2002, then data providers may randomly draw a number of 1000, which is kept secret, to shift dates. The genome-associated dates 2000, 2001 and 2002 are then shared as 3000, 3001 and 3002. While currently implausible, these translated dates are usable in phylodynamic analyses and preserve the distance between sampling dates. Once results are returned the data provider can internally account for the translation in any estimated ages, such as node ages or the tMRCA, by subtracting 1000. For example if the estimated age of the outbreak (taken as the tMRCA) was 5 years before the most recent sample, then the data provider can privately estimate the outbreak's onset as 1997 (2002 - 5), while those conducting the analysis externally can only estimate the relative age of 5 years. In the same way, intervals of transmission parameters such as $R_e$ can be placed with respect to the true sampling dates. Rates, such as growth or infection rates can also be accurately estimated via this method since these are not biased by shifting dates uniformly in time.

## Distributed computing

Approaches based on distributed computing, where data are analysed across remote servers, also offer promise for maximising data transparency and patient confidentiality. For example, [49] recently developed a method to estimate phylogenetic trees from private genome data using distributed computing and quantum crytographic protocols. Routine phylodynamic analysis for genomic surveillance may also benefit from adopting protocols from so-called swarm learning approaches that allow artificial intelligence models in precision medicine to be trained across distributed datasets (together comprising a swarm) [50]. Such approaches are in general complementary with hub-and-spoke networks, which are commonly used for storing sensitive pathogen genome data in national repositories [51]. We remain optimistic that future advances in distributed computing can eliminate the need for date-rounding in phylodynamic analysis.

## Supporting information

**S1 File.S1 Appendix.**
(PDF)

**Fig A in S1 File. Mean posterior variance for parameters of interest for each replicate of simulated data varying across date resolution.** Individual lines track mean posterior variance for each simulated dataset and boxplots are given to summarise variance in each condition at each date resolution. Rows correspond to individual parameters, columns correspond to simulation conditions (underlying parameters matching each empirical dataset), and colour corresponds to tree prior or reproductive number interval. (**A**) Mean posterior

variance in substitution rate across simulation scenarios. (**B**) Posterior variance in tMRCA, a measure of the age of the population driving the outbreak. (**C**) Posterior variance in reproductive number reproductive number.
(PDF)

**Fig B in S1 File. Desnsitrees (overlaid posterior trees) for empirical data with columns corresponding to pathogen under each combination of date resolution and tree prior.** For the H1N1 and SARS-CoV-2 treatments, Year resolution causes trees to collapse to instantaneous bursts.
(PDF)

**Fig C in S1 File. Adjusted phylodynamic likelihood against adjusted phylogenetic likelihood with panels corresponding to each simulation condition.** Points correspond to mean posterior likelihood for each simulated dataset under each simulation condition. Colour corresponds to date resolution. Likelihoods are adjusted by subtracting the mean phylodynamic or phylogenetic likelihood at day resolution from each the means under month and year resolution. Resulting points therefore show the difference phylodynamic and phylogenetic likelihoods due to date-rounding with the point $(0,0)$ representing likelihood at day resolution for each dataset. Month resolution generally results in smaller differences that Year resolution, suggesting coarser date resolution results in more perturbed likelihoods. There is also generally more error in phylodynamic likelihood than phylogenetic likelihood.
(PDF)

**Fig D in S1 File. Posterior distributions for the substitution rate, tMRCA, and $R_0$ for the H3N2 dataset.** The x-axis corresponds to the date resolution used in each re-analysis, and horizontal lines denote the median and 95% HPD bounds in each distribution. Overall, the H3N2 dataset is less sensitive to date rounding, but displays the same patterns of bias with decreasing date resolution as the comparable viral datasets of H1N1 and SARS-CoV-2 in the main text.
(PDF)

**Fig E in S1 File. The number of samples over time for each empirical dataset.** Date-rounding has the effect of moving each sampling within a month or year to the middle of that month or year ($15^{th}$ of the month or June $15^{th}$ of the year).
(PDF)

**Table A in S1 File. Mean posterior estimates of substitution rate and tMRCA for empirical data with 95% HPD in brackets.** The lower table gives mean posterior estimates of $R_\bullet$ for empirical data with 95% HPD in brackets.
(PDF)

## Author contributions

**Conceptualization:** Leo A. Featherstone.

**Data curation:** Leo A. Featherstone, Danielle J. Ingle.

**Formal analysis:** Leo A. Featherstone.

**Investigation:** Leo A. Featherstone.

**Methodology:** Leo A. Featherstone.

**Supervision:** Wytamma Wirth, Sebastian Duchene.

**Validation:** Leo A. Featherstone.

**Visualization:** Leo A. Featherstone.

**Writing – original draft:** Leo A. Featherstone.

**Writing – review & editing:** Leo A. Featherstone, Danielle J. Ingle, Wytamma Wirth, Sebastian Duchene.

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
