## [Decision Letter · Decision Letter 0]

25 Oct 2024

PCOMPBIOL-D-24-01565How does date-rounding affect phylodynamic inference for public health?PLOS Computational Biology Dear Dr. Leo, Thank you for submitting your manuscript to PLOS Computational Biology. After careful consideration, we feel that it has merit but does not fully meet PLOS Computational Biology's publication criteria as it currently stands. Therefore, we invite you to submit a revised version of the manuscript that addresses the points raised during the review process. Please submit your revised manuscript within 30 days Dec 25 2024 11:59PM. If you will need more time than this to complete your revisions, please reply to this message or contact the journal office at ploscompbiol@plos.org. Please include the following items when submitting your revised manuscript:* A rebuttal letter that responds to each point raised by the editor and reviewer(s). You should upload this letter as a separate file labeled 'Response to Reviewers'. This file does not need to include responses to formatting updates and technical items listed in the 'Journal Requirements' section below.* A marked-up copy of your manuscript that highlights changes made to the original version. You should upload this as a separate file labeled 'Revised Manuscript with Track Changes'.* An unmarked version of your revised paper without tracked changes. You should upload this as a separate file labeled 'Manuscript'. If you would like to make changes to your financial disclosure, competing interests statement, or data availability statement, please make these updates within the submission form at the time of resubmission. Guidelines for resubmitting your figure files are available below the reviewer comments at the end of this letter. We look forward to receiving your revised manuscript. Kind regards, Joel O. WertheimAcademic EditorPLOS Computational Biology Virginia PitzerSection EditorPLOS Computational Biology

Feilim Mac Gabhann

Editor-in-Chief

PLOS Computational Biology

Jason Papin

Editor-in-Chief

PLOS Computational Biology

 **Journal Requirements:** **Additional Editor Comments (if provided):** In addition to the reviewers’ comments and suggestions, I would also appreciate if you could address the following points. First, I believe the use of term ‘bias’ is inconsistent. The error is described as “unpredictable in its direction and compounds with decreasing date-resolution”, which sounds a lot like increasing variance to me. Bias, in contrast, would be that inferred dates are consistently younger when tip-date uncertainty is magnified [or inferred dates are older, but not both at the same time]. However, the directionality here appears to be highly model and organism dependent. Hence, it seems that tip-date uncertainty leads to greater variance and uncertainty, but not necessarily bias. Second, why does the Discussion directly name two pathogen nucleotide repositories (GISAID and Pathoplexus) and make no mention of other, larger governmental databases like GenBank, where GISAID and Pathoplexus currently import many sequences from?**Reviewers' comments:** Reviewer's Responses to Questions

**Comments to the Authors:**

Reviewer #1: Phylodynamic inference uses temporal information in the form of the sampling dates of the sequences to generate time-calibrated phylogenies from which tMRCAs, evolutionary substitution rates, and epidemiological parameters can be estimated. However, sampling month and day information are not always available, so strategies like rounding the sampling dates to the day or month are commonly used. Featherstone et al. present a comprehensive study of date-rounding's effect on the inference of important epidemiological and evolutionary parameters in a phylodynamic analysis, using both simulated and empirical datasets of viral and bacterial pathogens with varying evolutionary rates and sampling windows. The negative effects of rounding were more prominent for fast-evolving RNA viruses. The authors also argue that the bias introduced by date-rounding can be reduced (although not completely removed) when the unit of time used for the rounding (i.e., day or month) is shorter than the average time to observe a substitution in the sequences. Furthermore, they discuss some strategies to facilitate sharing complete sampling dates while protecting patient confidentiality.

The manuscript is well-structured and easy to read, the methods are clearly explained, and the results are discussed in detail. This study tackles one of the major issues in Phylodynamics, while also using relevant pathogens for the analyses. I have only some minor comments for this manuscript:

• Why wasn’t the coalescent prior used for the bacterial pathogens? Adding the reason would make the methods section clearer.

• I understand that it is recommended to avoid improper priors in a Bayesian phylogenetic analysis. What is the advantage of using an improper prior like 1/X over a Uniform with a very high upper bound (let’s say 100 million for the population size)?

• Line 200: delete every in “since every the target”

• Line 203: could the authors add a brief explanation of the origin parameter? Does it represent the age of the outbreak mentioned in line 186?

• Line 230: replace “birth-tree” by “birth-death tree”

• Line 271 remove duplicated words: At the year resolution there is there is

• In figure 3, panel B, the coalescent results are missing for SARS-CoV-2. Was this done on purpose? If yes, add the justification on the figure legend.

Reviewer #2: The manuscript entitled “How does date-rounding affect phylodynamic inference for public health?” presents an elegant analysis with a clear public health application. The authors investigate whether rounding of the sampling date introduces bias in phylodynamic inference, along with the magnitude and directionality of this bias based on several parameters. They also present a solution of how to avoid the rounding while preserving patient’s confidentiality. Overall, the article is well written and presented and I recommend it for publication. A few suggestions/questions that could help improve the manuscript include:

1. Is there a possibility to include an example of a viral rapidly evolving pathogen sampled over a longer period of time (a few years or decades)? I wonder if the same “clustering in time” issue will be a problem when there are more clusters that are distributed over time (i.e. samples collected over 48 months rather than 9). As we are moving into year 5 of COVID and year 40+ with HIV, it would be interesting to consider how the dates resolution affect phylodynamic inference of viral infections of public health concern sampled over a longer period of time.

2. It is not obvious to me why the authors insist on including the coalescent tree prior for SARS-CoV-2 dataset even though it violates the sampling proportion assumption (as they highlight in the text). Since this tree prior is already not included for the bacterial infections, I would consider removing.

3. Also, I understand that there is no effective “true value” for the empirical datasets, but maybe a range of “plausible values” based on the literature could be helpful for figure 3.

4. I suggest that the last paragraph in the Introduction to be moved to the Methods as it refers to the study setting and design.

5. Line 123-124 – did you mean “rounded to weeks”? If not, then unclear why Figure 1 refers to weeks and not days.

6. A few questions about the simulation setup that are not clear from the text: please explain how you define the proportion of sequenced cases in your simulations to meet the empirical datasets. Also, what did you consider to be the outbreak duration in your simulation?

7. Please describe the tree priors at the first time you refer to them (line 160).

8. Figure 2 – order legend in the same order as colors appear in the figure.

9. Line 271 and 395 typos.

**Have the authors made all data and (if applicable) computational code underlying the findings in their manuscript fully available?**

Reviewer #1: Yes

Reviewer #2: Yes

PLOS authors have the option to publish the peer review history of their article (what does this mean?). If published, this will include your full peer review and any attached files.

Reviewer #1: No

Reviewer #2: No

---

## [Decision Letter · Decision Letter 1]

21 Feb 2025

Dear Mr Leo,

We are pleased to inform you that your manuscript 'How does date-rounding affect phylodynamic inference for public health?' has been provisionally accepted for publication in PLOS Computational Biology.

Best regards,

Joel O. Wertheim

Academic Editor

PLOS Computational Biology

Thomas Leitner

Section Editor

PLOS Computational Biology

Reviewer's Responses to Questions

**Comments to the Authors:**

Reviewer #1: The authors have satisfactorily addressed all my comments, improving the quality and clarity of the manuscript. I have no further remarks.

Reviewer #2: I thank the authors for their careful consideration of my comments. I think the addition of the supplementary H3N2 dataset analysis strengthens the manuscript and their conclusions.

**Have the authors made all data and (if applicable) computational code underlying the findings in their manuscript fully available?**

Reviewer #1: Yes

Reviewer #2: Yes

PLOS authors have the option to publish the peer review history of their article (what does this mean?). If published, this will include your full peer review and any attached files.

Reviewer #1: No

Reviewer #2: No

---

## [Editor Report · Acceptance letter]

PCOMPBIOL-D-24-01565R1

How does date-rounding affect phylodynamic inference for public health?

Dear Dr Featherstone,

I am pleased to inform you that your manuscript has been formally accepted for publication in PLOS Computational Biology. Your manuscript is now with our production department and you will be notified of the publication date in due course.

With kind regards,

Anita Estes
